# Obesity, Inflammation, and Mortality in COVID-19: An Observational Study from the Public Health Care System of New York City

**DOI:** 10.3390/jcm11030622

**Published:** 2022-01-26

**Authors:** Leonidas Palaiodimos, Ryad Ali, Hugo O. Teo, Sahana Parthasarathy, Dimitrios Karamanis, Natalia Chamorro-Pareja, Damianos G. Kokkinidis, Sharanjit Kaur, Michail Kladas, Jeremy Sperling, Michael Chang, Kenneth Hupart, Colin Cha-Fong, Shankar Srinivasan, Preeti Kishore, Nichola Davis, Robert T. Faillace

**Affiliations:** 1NYC Health + Hospitals, New York, NY 10461, USA; hugo.teo@nychhc.org (H.O.T.); chamorrn1@nychhc.org (N.C.-P.); damiankokki@gmail.com (D.G.K.); kaurs31@nychhc.org (S.K.); kladasm@nychhc.org (M.K.); Jeremy.Sperling@nychhc.org (J.S.); changm12@nychhc.org (M.C.); Kenneth.Hupart@nychhc.org (K.H.); Colin.Cha-fong@nychhc.org (C.C.-F.); Preeti.Kishore@nychhc.org (P.K.); Nichola.Davis@nychhc.org (N.D.); Robert.Faillace@nychhc.org (R.T.F.); 2Jacobi Medical Center, Albert Einstein College of Medicine, Bronx, NY 10461, USA; 3Department of Health Informatics, Rutgers School of Health Professions, Newark, NJ 07107, USA; ryadali@shp.rutgers.edu (R.A.); dkaramanis@hotmail.com (D.K.); srinivsh@shp.rutgers.edu (S.S.); 4Division of General Internal Medicine, Icahn School of Medicine at Mount Sinai, New York, NY 10029, USA; 5Department of Population Health, NYU Grossman School of Medicine, New York, NY 10016, USA

**Keywords:** obesity, body mass index, BMI, COVID-19, New York, retrospective, observational, mortality, predictor, risk factor

## Abstract

Severe obesity increases the risk for negative outcomes in patients with coronavirus disease 2019 (COVID-19). Our objectives were to investigate the effect of BMI on in-hospital outcomes in our New York City Health and Hospitals’ ethnically diverse population, further explore this effect by age, sex, race/ethnicity, and timing of admission, and, given the relationship between COVID-19 and hyperinflammation, assess the concentrations of markers of systemic inflammation in different BMI groups. A retrospective study was conducted in hospitalized patients with COVID-19 in the public health care system of New York City from 1 March 2020 to 31 October 2020. A total of 8833 patients were included in this analysis (women: 3593, median age: 62 years). The median body mass index (BMI) was 27.9 kg/m^2^. Both overweight and obesity were independently associated with in-hospital death. The association of overweight and obesity with death appeared to be stronger in men, younger patients, and individuals of Hispanic ethnicity. We did not observe higher concentrations of inflammatory markers in patients with obesity as compared to those without obesity. In conclusion, overweight and obesity were independently associated with in-hospital death. Obesity was not associated with higher concentrations of inflammatory markers.

## 1. Introduction

There is strong and consistent evidence that obesity increases the risk for worse outcomes in patients with coronavirus disease 2019 (COVID-19) [1,2,3,4]. Severe obesity (body mass index (BMI) ≥ 35 kg/m^2^), in particular, has been associated with five-fold risk for admission in an intensive care unit (ICU) [5], a seven-fold risk for intubation and mechanical ventilation [6], and four-fold risk for in-hospital death [7]. Subgroup analyses of large cohort studies showed that this risk seems to be more pronounced in patients with obesity that are younger than 60 years and are men [4,8].

Specific racial and ethnic minority groups and individuals of low socioeconomic status have been disproportionally affected by COVID-19 [9,10,11,12,13,14]. Some of the factors that are responsible for the worse outcomes in these groups are a high housing density, limited health care access, and a higher prevalence of comorbidities known to be associated with severe COVID-19, such as hypertension, diabetes, and obesity [9,15]. As far as obesity is concerned, in the United States, Black and Hispanic adults have higher age-adjusted rates (49.6% and 44.8%, respectively) compared to White (42.2%) and Asian (17.4%) adults [16]. College graduates have a lower prevalence of obesity compared to those with less education [16].

Severe COVID-19 is strongly associated with hyperinflammation that is triggered by the severe acute respiratory syndrome coronavirus 2 (SARS-CoV-2) and is evident by higher levels of inflammatory markers and pro-inflammatory cytokines in the systemic circulation [17,18]. Patients with obesity have been demonstrated to have a dysregulated immune response and chronic low-grade inflammation accompanied by elevated pro-inflammatory cytokines, suggesting that these individuals may be more susceptible to hyperinflammation [19,20]. Therefore, there is particular interest in studying the inflammatory state and the possibly impaired immune response of individuals with obesity [21]. There is also a paucity of large studies evaluating the levels of inflammatory cells, inflammatory markers, and cytokines in patients with obesity and in different BMI ranges [22]. This is the case, especially, when it comes to patients with COVID-19 who are also of a lower socioeconomic status or belong to underrepresented minorities and diverse ethnic populations. Hence, there is a large unmet need for data showing how obesity affects their disease course and outcomes.

Our primary objective with this analysis was, therefore, to investigate the effect of BMI on in-hospital outcomes in a large cohort of patients with COVID-19 treated in the New York City public hospital system (NYC Health + Hospitals), which almost exclusively serves individuals of low socioeconomic status, most of whom belong to racial and ethnic minority groups. Our secondary objective was to further explore this effect by age, sex, race/ethnicity, and timing of admission. Our tertiary objective was to assess the concentrations of markers of systemic inflammation in different BMI groups.

## 2. Materials and Methods

### 2.1. Study Design, Study Setting, and Patient Population

We conducted a retrospective observation study on data collected at the eleven acute care hospitals of the New York City Health + Hospitals (NYC H + H) system, which is the largest public hospital system in the United States [23]. NYC H + H serves approximately one million individuals each year, the vast majority of whom are low-income (32% are uninsured and 35% are Medicaid beneficiaries) and/or belong to racial or ethnic minority groups (70% are people of color) [23]. Patients ≥ 18 years of age who presented to the emergency room and were admitted to any inpatient service, including the intensive care unit (ICU) with laboratory-confirmed COVID-19 from 1 March 2020 to 31 October 2020 were included. Laboratory-confirmed COVID-19 was defined as a SARS-CoV-2 positive result in real-time reverse transcriptase-polymerase chain reaction (RT-PCR) analysis of nasopharyngeal or nasal swab samples. We excluded patients who met any one of the following criteria: (i) patients < 18 years old; (ii) patients without laboratory-confirmed COVID-19; (iii) patients who were still hospitalized at the time of data collection; (iv) patients without recorded BMI at the time of the index hospitalization; (v) patients with extreme BMI values (BMI < 10 kg/m^2^, BMI > 60 kg/m^2^); (vi) women who were pregnant at the time of the index hospitalization. The study was approved by the Biomedical Research Alliance of New York Institutional Review Board with a waiver of informed consent (IRB #20-12-103-373). Data were fully de-identified and anonymized before the data was accessed and the IRB waived the requirement for informed consent.

### 2.2. Data Sources

Study data were obtained from electronic health records via appropriate diagnostic codes (Epic systems, Verona, WI, USA). The initial dataset was reviewed by two independent investigators for accuracy. Three pairs of additional independent investigators reviewed individual charts when clarifications were needed. The extracted data included age, sex, race, ethnicity, BMI, history of tobacco use, hypertension, hyperlipidemia, diabetes, coronary artery disease (CAD), heart failure, stroke, peripheral artery disease, atrial fibrillation, chronic obstructive pulmonary disease, asthma, chronic kidney disease (CKD) and end-stage renal disease (ESRD), liver cirrhosis, human immunodeficiency virus infection (HIV) or acquired immunodeficiency syndrome (AIDS), laboratory data including white blood cell count (WBC), neutrophil count, lymphocyte count, monocyte count, albumin, ferritin (electrochemiluminescence immunoassay (ECLIA) monoclonal ferritin-specific antibody (Roche Diagnostics)), lactate dehydrogenase (LDH), d-dimer, C-reactive protein (CRP) (particle enhanced immunoturbidimetric assay (Roche Diagnostics)), and interleukin-6 (IL-6) (Elecsys, Roche Diagnostics), and outcomes including invasive mechanical ventilation, admission to ICU, death, and hospital discharge. All laboratory tests refer to the first available results that were measured within 24 h from admission. The data were processed and analyzed without any personal identifiers to maintain patient confidentiality as per the Health Insurance Portability and Accountability Act (HIPAA).

### 2.3. Exposure of Interest and Outcomes

The primary exposure of interest was BMI. Patients were classified into five groups based on BMI: normal weight (BMI < 25 kg/m^2^), overweight (BMI 25 kg/m^2^ to <30 kg/m^2^), class I obesity (BMI 30 kg/m^2^ to <35 kg/m^2^), class II obesity (BMI 35 kg/m^2^ to <40 kg/m^2^), and class III obesity (BMI ≥ 40 kg/m^2^). No separate group for underweight (BMI ≤ 18.5 kg/m^2^) was set because a very low number of observations was expected in this group compared to the other groups, as well as the total number of included patients. The primary endpoint was in-hospital mortality. The secondary endpoints were invasive mechanical ventilation and admission to the ICU.

### 2.4. Statistical Analysis

ANOVA tests compared continuous variables, while chi-squared tests compared discrete variables. Continuous data are presented as a median value with the interquartile range (IQR) specified and categorical data are presented as absolute and relative frequencies.

Subgroup analysis of the association of inflammatory marker concentrations with BMI groups was conducted for sex, age (<65 years and ≥65 years), and survival status.

A stepwise logistic regression model identified baseline variables associated with in-hospital mortality, invasive mechanical ventilation, and admission to ICU. BMI < 25 kg/m^2^ was used as a reference in order to perform dichotomous comparisons with patients in the other BMI groups. Four multivariate models with different definitions of our variable of interest are presented for robustness: model A—BMI classified in five groups (<25 kg/m^2^, 25 to <30 kg/m^2^, 30 to <35 kg/m^2^, 35 to <40 kg/m^2^, ≥40 kg/m^2^), age, sex, and all available baseline characteristic variables; model B—BMI ≥30 kg/m^2^, age, sex, and all available baseline characteristic variables; model C—BMI ≥35 kg/m^2^, age, sex, and all available baseline characteristic variables; and model D—BMI ≥40 kg/m^2^, age, sex, and all available baseline characteristic variables. Age was treated as an ordinal variable (<50 years (reference), 50–64 years, 65–74 years, ≥75 years). Subgroup analyses for sex, age (<65 years and ≥65 years), race/ethnicity, and timing of hospitalization based on the results of logistic regression were conducted to further characterize the association of BMI with the outcome of mortality.

Another logistic regression model was used to identify inflammatory cells and markers associated with in-hospital mortality. Three different patient cohorts were utilized in this model; cohort A included 5068 patients for whom data were available for the inflammatory cells and markers of WBC count, neutrophil count, lymphocyte count, monocyte count, albumin, ferritin, LDH, and d-dimer; cohort B included 2461 patients for whom data were available for the inflammatory cells and markers of cohort A plus CRP; and cohort C included 665 patients for whom data were available for the inflammatory cells and markers of cohort B plus IL-6. Age, sex, and BMI were included in this logistic regression model. Age was treated as an ordinal variable (<50 years (reference), 50–64 years, 65–74 years, ≥75 years). BMI was analyzed as an ordinal variable (<25 kg/m^2^ (reference), 25 to <30 kg/m^2^, 30 to <35 kg/m^2^, 35 to <40 kg/m^2^, ≥40 kg/m^2^). Inflammatory cells and markers were treated as continuous variables. Subgroup analysis for BMI group (<25 kg/m^2^, 25 to <30 kg/m^2^, 30 to <35 kg/m^2^, 35 to < 40 kg/m^2^, ≥40 kg/m^2^) based on the results of logistic regression was conducted to further characterize the association of inflammatory cells and markers concentrations with the outcome of mortality. The areas under the curve (AUC) for inflammatory markers associated with mortality were also computed.

Results of logistic regression are given as the odds ratio (OR) with the 95% confidence interval (CI). The threshold of statistical significance was *p* < 0.05. All analyses were performed using SAS software (Release 9.4M6; SAS Institute, Cary, NC, USA).

## 3. Results

### 3.1. Descriptive Analyses

#### 3.1.1. Baseline Characteristics

In total, 8833 patients were included in this analysis, 3593 women (40.7%) and 5240 men (59.3%). Median BMI was 27.9 (IQR 24.3–32.6) kg/m^2^. BMI fell into the following groups: <25 kg/m^2^ = 2584 (29.3%), 25 to <30 kg/m^2^ = 2979 (34.0%), 30 to <35 kg/m^2^ = 1754 (20.0%), 35 to <40 kg/m^2^ = 833 (9.4%), ≥40 kg/m^2^ = 683 (8.0%). The median age of the entire cohort was 62 (IQR 49–74) years, with significant differences among the five BMI groups (<25 kg/m^2^: 68 (IQR 55–80); 25 to <30 kg/m^2^: 62 (IQR 49–74); 30 to <35 kg/m^2^: 59 (IQR 47–70); 35 to <40 kg/m^2^: 59 (IQR 46–68); ≥40 kg/m^2^: 55 (IQR 43–67); *p* < 0.001). A total of 38.9% of patients were of Hispanic ethnicity, 30.6% were non-Hispanic Black, 9.4% were non-Hispanic White, 5.0% were Asian, and 16.0% were of other or unknown race/ethnicity. Diabetes, hypertension, and hyperlipidemia were prevalent in 75.6%, 52.4%, and 23.7% of our patients, respectively. A total of 36.5% had history of stroke, 12.0% had CKD or ESRD, 10.3% had history of asthma, and 8.9% had history of CAD. Detailed baseline patient characteristics are presented in Table 1.

#### 3.1.2. Inflammatory Cells and Markers

Presentation of baseline concentrations of inflammatory cells and markers are presented in Table 2. Median baseline values of LDH, d-dimer, ferritin, CRP, IL-6, and neutrophil–lymphocyte ratios were significantly higher in men, patients ≥ 65 years, and deceased compared to women, patients < 65 years, and survivors, respectively (*p* < 0.001). Significant and consistent differences of baseline concentrations of inflammatory markers among different BMI groups were not noted (Figure 1, Figure 2 and Figure 3, detailed presentation in Appendix A).

#### 3.1.3. Outcomes

A total of 25.7% of our cohort died during hospitalization, with significantly higher rates among individuals with class II and class III obesity (BMI < 25 kg/m^2^: 24.9%, BMI 25 to <30 kg/m^2^: 25.0%, 30 to <35 kg/m^2^: 24.5%, 35 to <40 kg/m^2^: 28.3%, ≥40 kg/m^2^: 31.2%, *p* = 0.001). A J-shaped association between BMI and mortality rate was demonstrated (Figure 4). The mortality rates were 29.2% in the first patient quartile (3 March to 29 March 2020), 34.5% in the second (29 March to 8 April 2020), 29.0% in the third (8 April to 23 April 2020), and 9.0% in the fourth quartile (23 April to 31 October 2020). A total of 9.9% of patients received invasive mechanical ventilation and 20.3% were admitted to the ICU. Patients with class II and class III obesity had higher rates of invasive mechanical ventilation and admission to the ICU. The median length of stay for the total cohort was 6.7 days with non-significant differences among the BMI groups. In-hospital outcomes are presented in Table 3.

### 3.2. Logistic Regression Analyses

#### 3.2.1. Baseline Characteristics and in-Hospital Mortality, Invasive Mechanical Ventilation, and Admission to ICU

Univariate associations with in-hospital mortality, mechanical invasive ventilation, and admission to ICU were examined for the available baseline demographic and clinical characteristics and are presented in Table 4, Table 5 and Table 6, respectively. Smoking history was not included in these analyses due to significant amount of missing data.

In the multivariable analysis for in-hospital mortality, when BMI was analyzed as an ordinal variable in Model A, patients with BMI 25 to <30 kg/m^2^ (OR: 1.220; 95% CI: 1.068–1.393, *p* = 0.003), BMI 30 to <35 kg/m^2^ (OR: 1.494; 95% CI: 1.278–1.746, *p* < 0.001), BMI 35 to <40 kg/m^2^ (OR: 2.164; 95% CI: 1.780–2.632, *p* < 0.001), and BMI ≥ 40 kg/m^2^ (OR: 2.242; 95% CI: 1.855–2.710, *p* < 0.001) were found to have significantly higher likelihood for in-hospital death. When BMI was analyzed as a dichotomous variable with BMI < 25 kg/m^2^ as reference, BMI ≥ 30 kg/m^2^ (Model B), BMI ≥ 35 kg/m^2^ (Model C), and BMI ≥ 40 kg/m^2^ (Model D) were also found to have significant associations with higher in-hospital mortality. Older age, male sex, and history of stroke were also found to be independently associated with higher likelihood for in-hospital death, while hypertension, hyperlipidemia, diabetes, and heart failure were shown to have an inverse association with in-hospital mortality. This multivariate analysis is presented in Figure 5 (model A) and Table 4 (all models).

In the multivariable analysis for invasive mechanical ventilation, when BMI was analyzed as an ordinal variable in Model A, patients with BMI 25 to <30 kg/m^2^ (OR: 1.399; 95% CI: 1.043–1.877, *p* = 0.025), BMI 35 to <40 kg/m^2^ (OR: 1.656; 95% CI: 1.114–2.464, *p* = 0.013), and BMI ≥40 kg/m^2^ (OR: 1.794; 95% CI: 1.173–2.745, *p* = 0.007) were found to have significantly higher likelihood for invasive mechanical ventilation. The difference was not statically significant for BMI 30 to <35 kg/m^2^, but a clear trend was noted (OR: 1.338; 95% CI: 0.957–1.871, *p* = 0.088). When BMI was analyzed as a dichotomous variable with BMI <25 kg/m^2^ as reference, BMI ≥ 35 kg/m^2^ (Model C) was found to have significant association with invasive mechanical ventilation. This multivariate analysis is presented in Table 5. Sensitivity analysis for the outcome of mechanical invasive ventilation after excluding the patients that met the outcome of mortality is presented in Appendix A.

In the multivariable analysis for admission to ICU, when BMI was analyzed as an ordinal variable in Model A, patients with BMI 30 to <35 kg/m^2^ (OR: 1.418; 95% CI: 1.209–1.665, *p* < 0.001), BMI 35 to <40 kg/m^2^ (OR: 1.689; 95% CI: 1.386–2.058, *p* < 0.001), and BMI ≥ 40 kg/m^2^ (OR: 2.047; 95% CI: 1.660–2.524, *p* < 0.001) were found to have significantly higher likelihood for admission to ICU. The difference was not significant for BMI 25 to <30 kg/m^2^ (OR: 1.147; 95% CI: 0.995–1.323, *p* = 0.059), but a trend towards significance was revealed. When BMI was analyzed as a dichotomous variable with BMI < 25 kg/m^2^ as reference, BMI ≥ 30 kg/m^2^ (Model B), BMI ≥ 35 kg/m^2^ (Model C), and BMI ≥ 40 kg/m^2^ (Model D) all were found to have significant associations with higher likelihood for admission to ICU. This multivariate analysis is presented in Table 6. A sensitivity analysis for the outcome of mechanical invasive ventilation after excluding the patients that met the outcome of mortality is presented in Appendix A.

#### 3.2.2. Subgroup Analyses for in-Hospital Mortality

A subgroup analysis based on sex revealed that men, but not women, with BMI 25 to <30 kg/m^2^ and 30 to <35 kg/m^2^ were independently associated with higher in-hospital mortality. In the higher BMI groups, both men and women had significantly higher in-hospital mortality, but the associations were found to be stronger in men. This subgroup analysis is presented in Table 7.

A subgroup analysis based on age (<65 years and ≥65 years) revealed that all BMI groups above 25 kg/m^2^ of both age groups were associated with higher likelihood for in-hospital mortality except the BMI group 25 to <30 kg/m^2^ where the association did not reach statistical significance, but a clear trend was noted. Overall, the associations seemed to be stronger in patients <65 years old. This subgroup analysis is presented in Table 8.

A subgroup analysis based on the ethnic and racial background revealed that all BMI groups in patients of Hispanic ethnicity were significantly associated with higher likelihood for in-hospital mortality as compared to patients with normal BMI. In non-Hispanic Black patients, BMI group 35 to <40 kg/m^2^ and BMI ≥ 40 kg/m^2^ were significantly associated with mortality. In BMI groups 25 to <30 kg/m^2^ and 30 to <35 kg/m^2^ a similar trend was noted, but it did not reach statistical significance. In non-Hispanic White and Asian patients, only BMI ≥ 40 kg/m^2^ was significantly associated with mortality but the population of these groups were significantly smaller compared to Hispanic and non-Hispanic Black patients. This subgroup analysis is presented in Table 9.

A subgroup analysis for the timing of hospitalization was performed after the cohort was divided in four patient quartiles based on the day of admission (3/3–29/3/2020, 29/3–8/4/2020, 8/4–23/4/2020, 23/4–31/10/2020). In general, obesity was associated with higher in-hospital mortality in the first three patient quartiles with stronger associations noted among patients in the first quartile. No association was noted in patients that were admitted in the fourth quartile. This subgroup analysis is presented in Table 10. The univariate associations between time of admission (all admissions were classified in four and ten quantiles based on time of presentation) and in-hospital mortality was explored and is presented Appendix A. Overall, the likelihood of death was noted to be lower with later time of admission.

#### 3.2.3. Inflammatory Cells and Markers and in-Hospital Mortality

In the univariate analysis, WBC, neutrophil count, neutrophil-lymphocyte ratio, LDH, ferritin, d-dimer, CRP, and IL-6 were found to be positively associated with in-hospital mortality. On the other hand, monocyte count, albumin, and lymphocyte–CRP ratio were inversely associated with in-hospital mortality. No significant association between lymphocyte count and in-hospital mortality was observed. In the multivariate analysis adjusted for age, male sex, BMI, and the available inflammatory cells and markers, LDH, ferritin, d-dimer, IL-6, and albumin retained their significant associations with in-hospital mortality in all three cohorts. CRP and neutrophil–lymphocyte ratio retained significant associations with in-hospital mortality in cohorts A and B but not in the smaller cohort C (after addition of IL-6). These univariate and multivariate analyses are presented in Table 11. The AUC for inflammatory markers associated with mortality ranged from 0.5996 to 0.7274 and are presented in Appendix A. A subgroup analysis of the multivariate results based on BMI group is presented in Appendix A.

## 4. Discussion

Our study investigated the association of BMI with in-hospital outcomes in a cohort of 8833 patients admitted with COVID-19 in the largest public health care system of the United States. We examined how different age groups, sex, race/ethnicity, and hospital admission timing affected this association. The concentrations of inflammatory markers in various BMI groups were also assessed. We found that overweight and obesity were independently associated with in-hospital death, admission to ICU, and invasive mechanical ventilation with more significant associations observed in the higher BMI groups. The association of overweight and obesity with death appeared to be stronger in men, younger patients, and individuals of Hispanic ethnicity compared to women, older patients, and individuals of other racial/ethnic backgrounds. The association of obesity with in-hospital death was predominant in the first six weeks of the study, then it was attenuated over time and eventually lost its significance. We did not observe higher concentrations of inflammatory markers in patients with obesity. The concentrations of inflammatory markers were significantly higher in men, older patients, and those who died compared to women, younger patients, and survivors invariably observed in all BMI groups.

Obesity has been recognized as one of the most important risk factors for worse outcomes and death in patients with COVID-19 since the first wave of the pandemic [1]. Observational studies have clearly demonstrated the association between obesity or higher BMI with severe disease [5,24], invasive mechanical ventilation [6,25,26], and mortality [4,7,8,26,27]. The existing evidence is summarized in a large systematic review and meta-analysis from Huang et al. [28]. Data from thirty studies (45,650 patients) were pooled and synthesized, revealing that higher BMI was positively associated with mortality (OR 1.49; 95% CI: 1.20–1.85) [28]. Our findings are in accordance with the existing literature. We found obesity to be independently associated with almost 70% higher likelihood for death in hospitalized patients with COVID-19, with even higher risk in patients with severe obesity. All obesity classes were independently associated with higher risk for death, admission to ICU, and invasive mechanical ventilation compared to normal BMI group with increasing effect size estimates in higher obesity classes. It should be emphasized that our study is the first large cohort, to the best of our knowledge, to reveal that not only obesity, but also overweight (25 ≤ BMI < 30 kg/m^2^) was independently associated with higher odds for death, admission to ICU, and invasive mechanical ventilation. Therefore, individuals with overweight and obesity of any class should be considered as high-risk patients when it comes to vaccination prioritization, eligibility for neutralizing antibodies, and overall management.

It is not clear whether the high BMI-associated mortality risk in patients with COVID-19 is uniform across different patient populations or not. Severe obesity was found to be a risk factor for death in patients younger than 65 years but not among elderly in a cohort of 2466 hospitalized patients from Manhattan, New York by Anderson et al. [26]. Similar results were obtained from another large study of 6916 patients from California by Tartof et al., where high BMI was associated with substantial risk for death in younger adults (age ≤ 60 years) and men, but not in older adults (age > 60 years) and women [8]. In contrast to those two studies, our results indicate that men with obesity of any class and women with severe obesity have substantial risk for death, even if the relationship was found to be stronger in men and younger individuals. These differences between study findings can be attributed to the larger number of participants in our study and also by the significantly higher death rate compared to the Tartof et al. study.

Our subgroup analysis based on race/ethnicity suggested that patients of Hispanic ethnicity with overweight and obesity had higher likelihood of death compared to patients in other racial and ethnic groups. To the best of our knowledge, this is the first study to reveal this observation. It is not clear whether this is a generalizable finding or if it is unique to the specific local characteristics of our patient population and the characteristics of our sample (patients of Hispanic ethnicity had the largest representation in all BMI groups apart from BMI ≥ 40 kg/m^2^). Further studies are needed to explore potential differences in the role of obesity as a risk factor in different ethnic/racial groups.

Time of admission was associated with higher likelihood for death until early April 2020 when the number of hospitalizations reached its peak in NYC Health and Hospitals, and in New York overall [29], leading to significant pressure on the health care system [30]. Subsequently, progression in time was strongly associated with decreasing likelihood for death over the rest of the study period with 75% lower odds for death in last patient quartile compared to the first one. The latter finding was also observed by Tartof et al., where the mortality risk of the patients enrolled in the last week of that study was almost 70% less compared to patients presented in the first week [8]. Our admission timing-based BMI subgroup analysis demonstrated no association between higher BMI and death in the patients of the last quartile as opposed to the first three ones. We believe that the lower pressure applied in our health care system after April due to decrease in patient volume along with better understanding of COVID-19 pathophysiology and the identification of efficacious treatments [31,32] led to significant decrease in the mortality rates of patients with obesity over time. It is possible that the remarkably fewer death events in the last patient quartile led to attenuation and eventually minimization of the role of obesity as a risk factor for death in this subgroup of our cohort.

Hyper-inflammation triggered by SARS-CoV-2 is a prominent feature of severe COVID-19 [33] and is characterized by high serum concentrations of inflammatory markers, inflammatory cytokines, and chemokines [17,18]. Increased levels of WBC, neutrophils, CRP, ferritin, LDH, d-dimer, and IL-6, as well as decreased levels of lymphocytes, monocytes, and albumin have been associated with severe COVID-19 and worse outcomes [34,35,36,37,38,39,40]. Systemic inflammation as indicated by these markers has not been adequately explored in COVID-19 patients with obesity, a population that suffers from dysregulated immune response and low-grade inflammation at baseline [19,20]. Metabolic inflammation has been proposed as a possible mechanism to explain the higher likelihood for negative outcomes that patients with COVID-19 and obesity have [41]. Therefore, we performed a comprehensive analysis of the serum concentrations of inflammatory markers on presentation in various BMI groups. We observed that the concentrations of CRP, ferritin, LDH, d-dimer, and IL-6, and neutrophil-to-lymphocyte ratio were significantly higher in patients who died compared to survivors, which is a finding consistent with the existing literature [42,43,44,45,46,47]. Men and older patients were found to have higher concentrations of CRP, ferritin, LDH, IL-6, and neutrophil-to-lymphocyte ratio compared to women and younger patients, which is in line with the known higher risk for worse outcomes in men and elderly [1]. Our subgroup analysis based on BMI revealed that these differences between deceased and survivors, men and women, older and younger patients were observed across all BMI groups indicating that while systemic inflammation plays an important role in the outcomes of patients with COVID-19 this is independent to rising BMI. Contrary to our hypothesis, we did not observe higher concentrations of inflammatory markers in patients with obesity suggesting that predisposition to hyperinflammation may not be the main pathogenetic factor behind worse outcomes in this population. Higher expression of angiotensin-converting enzyme 2 in visceral adipose tissue [48,49] may lead to higher viral load and ectopic fat tissue local inflammation in patients with obesity. Additionally, there is a high prevalence of vitamin D deficiency in patients with obesity [50], which has been described as a possible risk factor for worse outcomes in patients with COVID-19 [51]. Moreover, the predisposition to thromboembolism that patients with obesity have [52], which is also a common manifestation in COVID-19 [53], and the suboptimal mechanical properties of the lungs and chest wall in patients with severe obesity [54] are other pathogenetic mechanisms that can explain the worse outcomes of patients with obesity and COVID-19.

To our knowledge, our study is the largest to date assessing the differential impact of overweight and obesity in the in-hospital outcomes of patients with COVID-19 of different sexes, age groups, racial/ethnic backgrounds, and timing of hospital admission. This study also offers the most comprehensive evaluation of the associations of BMI with inflammatory markers in patients with COVID-19. The majority of our patients come from racial/ethnic groups that have been significantly affected by the pandemic and are largely underrepresented in the existing literature. We should acknowledge that our study has several limitations. First, this was a retrospective cohort involving electronic medical records, hence, there are risks related to observational bias and unmeasured confounding. Second, our cohort is unique in that patients mostly belong to low socio-economic strata; therefore, our findings cannot be easily generalized to patient populations with other characteristics.

## 5. Conclusions

In conclusion, in this large cohort of hospitalized patients with COVID-19 in a public health care system, we observed that overweight and obesity were associated with in-hospital death, admission to ICU, and invasive mechanical ventilation after adjusting for obesity-related potentially confounding factors. The association of overweight and obesity with death appeared to be stronger in men, younger patients, and individuals of Hispanic ethnicity. Patients with obesity did not have higher concentrations of inflammatory markers on presentation. Not only patients with severe obesity but also those with overweight should be highly encouraged to receive full vaccination regimen against SARS-CoV-2 and they should be prioritized to receive neutralizing antibodies or antivirals as early treatments. While systemic hyper-inflammation appears to play an important role in the adverse outcomes in patients with COVID-19, this effect appears to be independent of BMI, further studies are needed to explore additional mechanisms and possible therapeutic targets.

## Figures and Tables

**Figure 1 jcm-11-00622-f001:**
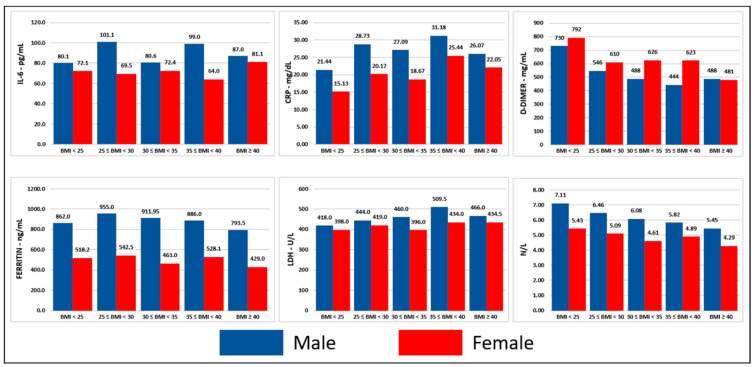
Inflammatory markers stratified per BMI group and survival status. Notes: (1) This figure was created based on the data that are presented in detail on Appendix A, (2) the values correspond to the median of the first available individual values that were obtained within 24 h from presentation, (3) BMI in kg/m^2^. Abbreviations: BMI = body mass index, ng = nanogram, mg = milligram, L = liter, dL = deciliter, mL = milliliter, U = unit, CRP = C-reactive protein, IL-6 = interleukin-6, LDH = lactic dehydrogenase, N/L = neutrophil–lymphocyte ratio.

**Figure 2 jcm-11-00622-f002:**
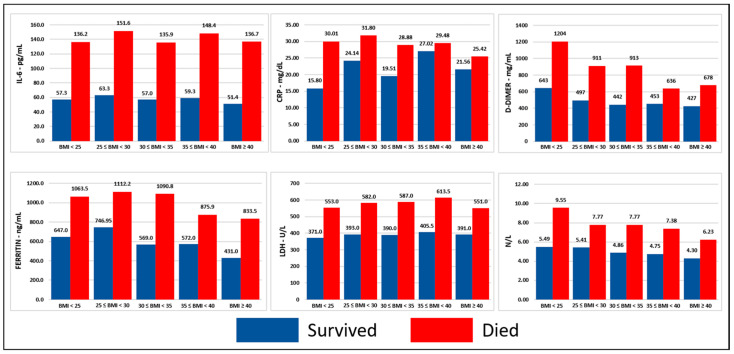
Inflammatory markers stratified per BMI group and sex. Notes: (1) This figure was created based on the data that are presented in detail on Appendix A, (2) the values correspond to the median of the first available individual values that were obtained within 24 h from presentation, (3) BMI in kg/m^2^. Abbreviations: BMI = body mass index, ng = nanogram, mg = milligram, L = liter, dL = deciliter, mL = milliliter, U = unit, CRP = C-reactive protein, IL-6 = interleukin-6, LDH = lactic dehydrogenase, N/L = neutrophil–lymphocyte ratio.

**Figure 3 jcm-11-00622-f003:**
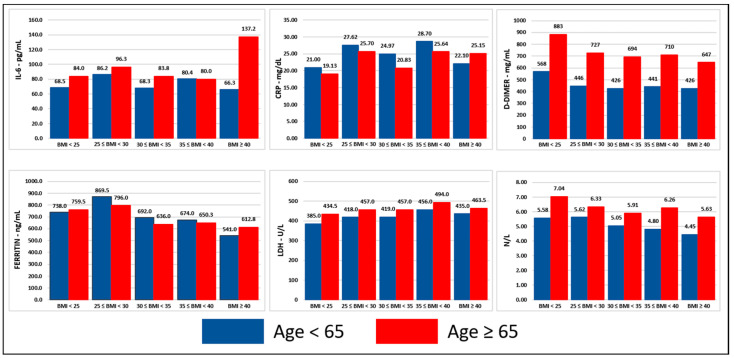
Inflammatory markers stratified per BMI group and age. Notes: (1) This figure was created based on the data that are presented in detail on Appendix A, (2) the values correspond to the median of the first available individual values that were obtained within 24 h from presentation, (3) BMI in kg/m^2^. Abbreviations: BMI = body mass index, ng = nanogram, mg = milligram, L = liter, dL = deciliter, mL = milliliter, U = unit, CRP = C-reactive protein, IL-6 = interleukin-6, LDH = lactic dehydrogenase, N/L = neutrophil–lymphocyte ratio.

**Figure 4 jcm-11-00622-f004:**
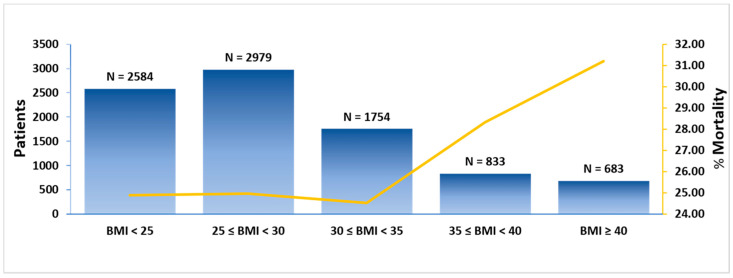
Mortality rate per BMI group. Notes: (1) BMI in kg/m^2^, (2) left vertical axis refers to number of patients, (3) right; vertical axis refers to mortality rate. Abbreviations: BMI = body mass index, kg = kilogram, m = meter.

**Figure 5 jcm-11-00622-f005:**
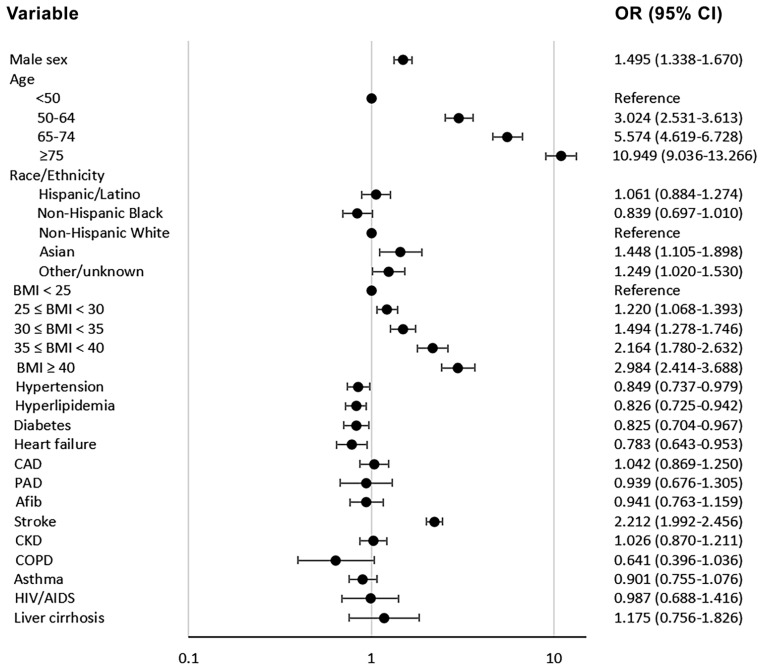
Forest plot depicting the results of the multivariate logistic regression analysis for the outcome of mortality (Model A). Notes: (1) BMI in kg/m^2^, (2) age in years, (3) Model A includes the variables that were found to have significant univariate association and BMI as ordinal variable: BMI < 25, 25 ≤ BMI < 30, 30 ≤ BMI < 35, 35 ≤ BMI < 40, and BMI ≥ 40. Abbreviations: BMI = body mass index, OR = odds ratio, CI = confidence interval, kg = kilogram, m = meter, CAD = coronary artery disease, PAD = peripheral artery disease, A fib = atrial fibrillation, CKD = chronic kidney disease, COPD = chronic obstructive pulmonary disease, HIV = human immunodeficiency, AIDS = acquired immunodeficiency syndrome.

**Table 1 jcm-11-00622-t001:** Baseline patient characteristics.

Characteristic	All Patients	BMI Group				
	*n* = 8833	BMI < 25	25 ≤ BMI < 30	30 ≤ BMI < 35	35 ≤ BMI < 40	BMI ≥ 40
		*n* = 2584	*n* = 2979	*n* = 1754	*n* = 833	*n* = 683
Male sex—no. (%)	5240 (59.32)	1695 (65.6)	1917 (64.35)	950 (54.16)	385 (46.22)	293 (42.9)
Age—years						
Median (IQR)	62 (49–74)	68 (55–80)	62 (49–74)	59 (47–70)	59 (46–68)	55 (43–67)
Distribution—no. (%)						
<50	2270 (25.7)	490 (18.96)	762 (25.58)	518 (29.53)	243 (29.17)	257 (37.63)
50–64	2601 (29.45)	606 (23.45)	881 (29.57)	580 (33.07)	309 (37.09)	225 (32.94)
65–74	1863 (21.09)	569 (22.02)	630 (21.15)	361 (20.58)	168 (20.17)	135 (19.77)
≥75	2099 (23.76)	919 (35.57)	706 (23.7)	295 (16.82)	113 (13.57)	66 (9.66)
Race/ethnicity—no. (%)						
Hispanic/Latino	3437 (38.91)	870 (33.67)	1255 (42.13)	767 (43.73)	333 (39.98)	212 (31.04)
Non-Hispanic Black	2702 (30.59)	794 (30.73)	800 (26.85)	520 (29.65)	296 (35.53)	292 (42.75)
Non-Hispanic White	832 (9.42)	306 (11.84)	256 (8.59)	147 (8.38)	62 (7.44)	61 (8.93)
Asian	445 (5.04)	190 (7.35)	177 (5.94)	50 (2.85)	19 (2.28)	9 (1.32)
Other/Unknown	1417 (16.04)	424 (16.41)	491 (16.48)	270 (15.39)	123 (14.77)	109 (15.96)
Smoking—no./total no. (%)						
Never Smoked	3063 (34.68)	769 (29.76)	1063 (35.68)	651 (37.12)	319 (38.3)	261 (38.21)
Active or former smoker	882 (9.99)	290 (11.22)	292 (9.8)	160 (9.12)	78 (9.36)	55 (8.05)
Unknown	4888 (55.34)	1525 (59.02)	1624 (54.51)	943 (53.76)	436 (52.34)	367 (53.73)
Comorbidities—no. (%)						
Hypertension	4632 (52.44)	1351 (52.28)	1538 (51.63)	884 (50.4)	467 (56.06)	392 (57.39)
Hyperlipidemia	2089 (23.65)	597 (23.1)	701 (23.53)	415 (23.66)	241 (28.93)	135 (19.77)
Diabetes	6674 (75.56)	2030 (78.56)	2175 (73.01)	1288 (73.43)	647 (77.67)	534 (78.18)
Heart failure	754 (8.54)	268 (10.37)	211 (7.08)	127 (7.24)	74 (8.88)	74 (10.83)
CAD	788 (8.92)	262 (10.14)	279 (9.37)	140 (7.98)	63 (7.56)	44 (6.44)
PAD	215 (2.43)	83 (3.21)	72 (2.42)	33 (1.88)	18 (2.16)	9 (1.32)
A fib	537 (6.08)	167 (6.46)	180 (6.04)	101 (5.76)	51 (6.12)	38 (5.56)
Stroke	3222 (36.48)	919 (35.57)	1136 (38.13)	616 (35.12)	298 (35.77)	253 (37.04)
CKD	1063 (12.03)	387 (14.98)	339 (11.38)	169 (9.64)	97 (11.64)	71 (10.4)
COPD	117 (1.32)	60 (2.32)	28 (0.94)	15 (0.86)	8 (0.96)	6 (0.88)
Asthma	911 (10.31)	222 (8.59)	224 (7.52)	184 (10.49)	127 (15.25)	154 (22.55)
HIV/AIDS	212 (2.4)	84 (3.25)	55 (1.85)	39 (2.22)	12 (1.44)	22 (3.22)
Liver cirrhosis	138 (1.56)	42 (1.63)	56 (1.88)	26 (1.48)	9 (1.08)	5 (0.73)

Notes: BMI in kg/m^2^. Abbreviations and symbols: BMI = body mass index; IQR = interquartile range; no. = number; kg = kilogram; m = meter; CAD = coronary artery disease; PAD = peripheral artery disease; A fib = atrial fibrillation; BMI: body mass index; CKD: chronic kidney disease; COPD = chronic obstructive pulmonary disease; HIV: human immunodeficiency virus infection; AIDS: acquired immunodeficiency syndrome.

**Table 2 jcm-11-00622-t002:** Baseline inflammatory markers.

Marker	All Patients		BMI Group									
			BMI < 25		25 ≤ BMI < 30		30 ≤ BMI < 35		35 ≤ BMI < 40		BMI ≥ 40	
	No.	Median (IQR)	No. (%)	Median (IQR)	No. (%)	Median (IQR)	No. (%)	Median (IQR)	No. (%)	Median (IQR)	No. (%)	Median (IQR)
WBC—10^3^/μL	8795	7.90 (5.79–11.01)	2574 (29.27)	8.13 (5.73–11.69)	2963 (33.69)	7.88 (5.83–10.77)	1745 (19.84)	7.78 (5.91–10.77)	832 (9.46)	7.89 (5.67–10.85)	681 (7.74)	7.53 (5.72–10.25)
Neutrophils—10^3^/μL	8795	5.94 (4.03–8.89)	2574 (29.27)	6.24 (3.95–9.72)	2963 (33.69)	5.96 (4.09–8.79)	1745 (19.84)	5.86 (4.13–8.46)	832 (9.46)	5.81 (3.99–8.58)	681 (7.74)	5.58 (3.86–8.04)
Monocytes—10^3^/μL	8795	0.51 (0.35–0.73)	2574 (29.27)	0.53 (0.36–0.75)	2963 (33.69)	0.50 (0.35–0.72)	1745 (19.84)	0.49 (0.34–0.71)	832 (9.46)	0.50 (0.33–0.72)	681 (7.74)	0.51 (0.34–0.72)
Lymphocytes—10^3^/μL	8795	1.03 (0.72–1.45)	2574 (29.27)	0.96 (0.66–1.39)	2963 (33.69)	1.01 (0.72–1.42)	1745 (19.84)	1.08 (0.77–1.53)	832 (9.46)	1.12 (0.78–1.49)	681 (7.74)	1.12 (0.81–1.57)
LDH—U/L	6476	432.0 (310.0–611.0)	1822 (28.13)	410.5 (289.0–580.0)	2176 (33.6)	434.0 (316.5–612.5)	1313 (20.27)	434.0 (317.0–615.0)	648 (10.01)	465.0 (331.0–666.0)	517 (7.98)	446.0 (334.0–626.0)
Ferritin—ng/mL	6285	740.0 (337.0–1447.0)	1790 (28.48)	744.8 (337.5–1549.0)	2085 (33.17)	835.2 (395.0–1544.0)	1269 (20.19)	668.0 (308.0–1366.1)	638 (10.15)	659.0 (318.0–1267.6)	503 (8)	574.0 (253.0–1157.0)
CRP—mg/dL	3114	24.22 (8.40–92.80)	897 (28.81)	19.57 (6.56–86.10)	997 (32.02)	26.82 (10.09–111.81)	614 (19.72)	23.01 (8.61–97.15)	327 (10.5)	28.44 (8.50–95.10)	279 (8.96)	22.99 (9.51–69.10)
Interleukin 6—pg/mL	2237	82.10 (35.50–174.30)	596 (26.64)	77.45 (33.55–175.15)	720 (32.19)	91.25 (39.60–191.25)	472 (21.1)	76.50 (34.55–156.85)	248 (11.09)	80.20 (36.30–188.15)	201 (8.99)	85.10 (32.30–164.00)
Albumin—g/dL	8357	3.70 (3.40–4.10)	2455 (29.38)	3.60 (3.20–4.00)	2809 (33.61)	3.80 (3.40–4.10)	1657 (19.83)	3.80 (3.40–4.10)	788 (9.43)	3.70 (3.40–4.00)	648 (7.75)	3.70 (3.40–4.00)
D-dimer—mg/mL	6471	598.0 (304.0–1715.0)	1844 (28.5)	766.0 (365.5–2371.5)	2197 (33.95)	565.0 (299.0–1357.0)	1286 (19.87)	528.0 (279.0–1333.0)	624 (9.64)	523.5 (287.0–1105.5)	520 (8.04)	484.0 (261.0–1079.0)
N/L	8795	5.74 (3.40–9.93)	2574 (29.27)	6.37 (3.53–11.33)	2963 (33.69)	5.98 (3.53–9.97)	1745 (19.84)	5.35 (3.30–8.99)	832 (9.46)	5.26 (3.21–8.86)	681 (7.74)	4.77 (3.01–7.81)
L/CRP	3109	3.98 (1.07–12.32)	896 (28.82)	4.64 (1.05–16.17)	995 (32)	3.46 (0.91–10.17)	613 (19.72)	4.11 (1.10–12.81)	327 (10.52)	3.14 (1.22–13.75)	278 (8.94)	4.61 (1.69–11.61)

Notes: (1) The first available laboratory values after presentation were considered in this analysis if within 24 h from presentation, (2) BMI in kg/m^2^. Abbreviations: BMI = body mass index, IQR = interquartile range, no. = number, g = gram, ng = nanogram, μg = microgram, mg = milligram, L = liter, μL = microliter, dL = deciliter, mL = milliliter, U = unit, WBC = white blood cell count, LDH = lactic dehydrogenase, CRP = C-reactive protein, N/L = neutrophils/lymphocytes, L/CRP = lymphocytes/C-reactive protein.

**Table 3 jcm-11-00622-t003:** Descriptive presentation of in-hospital outcomes.

Outcome	All Patients	BMI Group				
		BMI < 25	25 ≤ BMI < 30	30 ≤ BMI < 35	35 ≤ BMI < 40	BMI ≥ 40
	*n* = 8833	*n* = 2584	*n* = 2979	*n* = 1754	*n* = 833	*n* = 683
	No. (%)	No. (%)	No. (%)	No. (%)	No. (%)	No. (%)
Intubation	871 (9.86)	201 (7.78)	300 (10.07)	174 (9.92)	93 (11.16)	103 (15.08)
ICU	1796 (20.33)	442 (17.11)	580 (19.47)	387 (22.06)	203 (24.37)	184 (26.94)
Mortality	2266 (25.65)	643 (24.88)	744 (24.97)	430 (24.52)	236 (28.33)	213 (31.19)
with intubation ^2^	586 (25.86)	128 (19.91)	200 (26.88)	113 (26.28)	71 (30.08)	74 (34.74)
without intubation ^2^	1680 (74.14)	515 (80.09)	544 (73.12)	317 (73.72)	165 (69.92)	139 (65.26)
with ICU ^2^	938 (41.39)	179 (27.84)	302 (40.59)	222 (51.63)	120 (50.85)	115 (53.99)
without ICU ^2^	1328 (58.61)	464 (72.16)	442 (59.41)	208 (48.37)	116 (49.15)	98 (46.01)
1st quartile (3/3–29/3/2020)	644/2209 (29.15)	142/519 (27.36)	206/775 (26.58)	141/495 (28.48)	70/213 (32.86)	85/207 (41.06)
2nd quartile (29/3–8/4/2020)	783/2208 (35.46)	218/592 (36.82)	272/779 (34.92)	136/412 (33.01)	91/254 (35.83)	66/171 (38.60)
3rd quartile (8/4–23/4/2020)	640/2208 (28.99)	203/688 (29.51)	213/737 (28.90)	119/416 (28.61)	55/198 (27.78)	50/169 (29.59)
4th quartile 23/4–31/10/2020)	199/2208 (9.01)	80/785 (10.19)	53/688 (7.70)	34/431 (7.89)	20/168 (11.90)	12/136 (8.82)

Notes: (1) The outcomes are presented as no. (%), (2) ^2^—The relative frequencies refer to the subgroup of patients that died, (3) BMI in kg/m^2^, (4) the total cohort of 8833 patients was divided in four equal quartile cohorts based on the day of admission (1st quartile: 3/3–3/29/2020, 2nd quartile: 3/29–4/8/2020, 3rd quartile: 4/8–4/23/2020, 4th quartile: 4/23–10/31/2020). The first quartile had 2209 patients and the other three quartiles had 2208 patients. Abbreviations: BMI = body mass index, no. = number, ICU = admission to intensive care unit.

**Table 4 jcm-11-00622-t004:** Baseline patient characteristics: univariate and multivariate logistic regression analysis for the outcome of mortality.

Variable	Univariate Analysis	Multivariate Analysis			
		Model A	Model B	Model C	Model D
		*n* = 8833	*n* = 8833	*n* = 8833	*n* = 8833
	OR, 95% CI, *p*-Value	OR, 95% CI, *p*-Value	OR, 95% CI, *p*-Value	OR, 95% CI, *p*-Value	OR, 95% CI, *p*-Value
Male sex	1.192 (1.081–1.315) *p* < 0.001	1.495 (1.338–1.670) *p* < 0.001	1.450 (1.299–1.619) *p* < 0.001	1.451 (1.300–1.620) *p* < 0.001	1.401 (1.257–1.562) *p* < 0.001
Age (All)	1.045 (1.042–1.048) *p* < 0.001				
<50 (reference)	*	*	*	*	*
50–64	2.676 (2.259–3.170) *p* < 0.001	3.024 (2.531–3.613) *p* < 0.001	2.930 (2.455–3.498) *p* < 0.001	2.976 (2.492–3.554) *p* < 0.001	2.987 (2.501–3.566) *p* < 0.001
65–74	4.466 (3.761–5.303) *p* < 0.001	5.574 (4.619–6.728) *p* < 0.001	5.288 (4.387–6.373) *p* < 0.001	5.357 (4.443–6.459) *p* < 0.001	5.224 (4.335–6.295) *p* < 0.001
≥75	7.358 (6.234–8.684) *p* < 0.001	10.949 (9.036–13.266) *p* < 0.001	10.001 (8.281–12.078) *p* < 0.001	10.022 (8.300–12.101) *p* < 0.001	9.488 (7.871–11.438) *p* < 0.001
Ethnicity/Race					
Hispanic/Latino	0.597 (0.507–0.703) *p* < 0.001	1.061 (0.884–1.274) *p* = 0.522	1.048 (0.874–1.257) *p* = 0.611	1.071 (0.893–1.285) *p* = 0.457	1.073 (0.896–1.286) *p* = 0.443
Non-Hispanic Black	0.617 (0.521–0.730) *p* < 0.001	0.839 (0.697–1.010) *p* = 0.063	0.845 (0.702–1.016) *p* = 0.073	0.842 (0.700–1.013) *p* = 0.069	0.849 (0.706–1.020) *p* = 0.080
Non-Hispanic White	*	*	*	*	*
Asian	0.877 (0.685–1.123) *p* = 0.297	1.448 (1.105–1.898) *p* = 0.007	1.408 (1.076–1.843) *p* = 0.013	1.383 (1.057–1.810) *p* = 0.018	1.340 (1.025–1.752) *p* = 0.032
Other/unknown	0.792 (0.658–0.952) *p* = 0.013	1.249 (1.020–1.530) *p* = 0.031	1.245 (1.018–1.523) *p* = 0.033	1.247 (1.018–1.526) *p* = 0.033	1.246 (1.019–1.524) *p* = 0.032
BMI < 25 (reference)	*	*			
25 ≤ BMI < 30	1.005 (0.890–1.135) *p* = 0.938	1.220 (1.068–1.393) *p* = 0.003			
30 ≤ BMI < 35	0.980 (0.852–1.128) *p* = 0.783	1.494 (1.278–1.746) *p* < 0.001			
35 ≤ BMI < 40	1.193 (1.002–1.422) *p* = 0.048	2.164 (1.780–2.632) *p* < 0.001			
BMI ≥ 30	1.107 (1.003–1.221) *p* = 0.043		1.672 (1.495–1.870) *p* < 0.001		
BMI ≥ 35	1.274 (1.127–1.440) *p* < 0.001			2.046 (1.781–2.351) *p* < 0.001	
BMI ≥ 40	1.346 (1.136–1.594) *p* < 0.001	2.984 (2.414–3.688) *p* < 0.001			2.242 (1.855–2.710) *p* < 0.001
Hypertension	1.303 (1.184–1.435) *p* < 0.001	0.849 (0.737–0.979) *p* = 0.024	0.871 (0.757–1.003) *p* = 0.056	0.873 (0.758–1.006) *p* = 0.060	0.882 (0.766–1.015) *p* = 0.080
Hyperlipidemia	1.044 (0.934–1.167) *p* = 0.450	0.826 (0.725–0.942) *p* = 0.004	0.822 (0.721–0.936) *p* = 0.003	0.827 (0.726–0.941) *p* = 0.004	0.843 (0.741–0.960) *p* = 0.010
Diabetes	1.113 (0.994–1.246) *p* = 0.063	0.825 (0.704–0.967) *p* = 0.017	0.821 (0.701–0.962) *p* = 0.015	0.810 (0.691–0.949) *p* = 0.009	0.810 (0.692–0.948) *p* = 0.009
Heart failure	1.098 (0.929–1.299) *p* = 0.273	0.783 (0.643–0.953) *p* = 0.015	0.795 (0.654–0.967) *p* = 0.022	0.784 (0.644–0.953) *p* = 0.015	0.787 (0.648–0.957) *p* = 0.016
CAD	1.445 (1.235–1.691) *p* < 0.001	1.042 (0.869–1.250) *p* = 0.657	1.031 (0.860–1.236) *p* = 0.742	1.047 (0.873–1.255) *p* = 0.622	1.038 (0.866–1.245) *p* = 0.683
PAD	1.099 (0.811–1.488) *p* = 0.544	0.939 (0.676–1.305) *p* = 0.710	0.925 (0.667–1.285) *p* = 0.643	0.921 (0.664–1.279) *p* = 0.624	0.918 (0.661–1.273) *p* = 0.607
A fib	1.515 (1.258–1.824) *p* < 0.001	0.941 (0.763–1.159) *p* = 0.564	0.949 (0.771–1.168) *p* = 0.620	0.957 (0.778–1.178) *p* = 0.680	0.969 (0.788–1.193) *p* = 0.768
Stroke	2.462 (2.233–2.715) *p* < 0.001	2.212 (1.992–2.456) *p* < 0.001	2.218 (1.998–2.461) *p* < 0.001	2.221 (2.001–2.465) *p* < 0.001	2.210 (1.992–2.452) *p* < 0.001
CKD	1.100 (0.952–1.271) *p* = 0.195	1.026 (0.870–1.211) *p* = 0.757	1.023 (0.867–1.206) *p* = 0.789	1.008 (0.855–1.188) *p* = 0.925	1.007 (0.855–1.187) *p* = 0.931
COPD	0.745 (0.475–1.170) *p* = 0.202	0.641 (0.396–1.036) *p* = 0.070	0.623 (0.386–1.006) *p* = 0.053	0.616 (0.382–0.993) *p* = 0.047	0.610 (0.378–0.983) *p* = 0.043
Asthma	0.866 (0.737–1.019) *p* = 0.082	0.901 (0.755–1.076) *p* = 0.250	0.947 (0.795–1.129) *p* = 0.545	0.916 (0.768–1.093) *p* = 0.332	0.930 (0.780–1.109) *p* = 0.420
HIV/AIDS	0.732 (0.522–1.027) *p* = 0.071	0.987 (0.688–1.416) *p* = 0.942	0.968 (0.675–1.389) *p* = 0.862	0.974 (0.680–1.396) *p* = 0.887	0.937 (0.655–1.342) *p* = 0.724
Liver cirrhosis	0.802 (0.534–1.206) *p* = 0.290	1.175 (0.756–1.826) *p* = 0.473	1.165 (0.751 1.805) *p* = 0.496	1.161 (0.748–1.802) *p* = 0.505	1.142 (0.736–1.772) *p* = 0.555

Notes: (1) BMI in kg/m^2^, (2) age in years, (3) Model A includes the variables that were found to have significant univariate association and BMI as ordinal variable: BMI < 25, 25 ≤ BMI < 30, 30 ≤ BMI < 35, 35 ≤ BMI < 40, and BMI ≥ 40, (4) Model B includes the variables that were found to have significant univariate association and BMI as dichotomous variable: BMI < 30 and BMI ≥ 30, (5) Model C includes the variables that were found to have significant univariate association and BMI as dichotomous variable: BMI < 35 and BMI ≥ 35, (6) Model D includes the variables that were found to have significant univariate association and BMI as dichotomous variable: BMI < 40 and BMI ≥ 40. Abbreviations: BMI = body mass index, OR = odds ratio, CI = confidence interval, kg = kilogram, m = meter, CAD = coronary artery disease, PAD = peripheral artery disease, A fib = atrial fibrillation CKD = chronic kidney disease, COPD = chronic obstructive pulmonary disease, HIV = human immunodeficiency, AIDS = acquired immunodeficiency syndrome, * = reference group.

**Table 5 jcm-11-00622-t005:** Baseline patient characteristics: univariate and multivariate logistic regression analysis for the outcome of invasive mechanical ventilation.

Variable	Univariate Analysis	Multivariate Analysis			
		Model A	Model B	Model C	Model D
		*n* = 8833	*n* = 8833	*n* = 8833	*n* = 8833
	OR, 95% CI, *p*-Value	OR, 95% CI, *p*-Value	OR, 95% CI, *p*-Value	OR, 95% CI, *p*-Value	OR, 95% CI, *p*-Value
Male sex	1.512 (1.303–1.756) *p* < 0.001	1.553 (1.328–1.816) *p* < 0.001	1.509 (1.292–1.764) *p* < 0.001	1.515 (1.297–1.771) *p* < 0.001	1.485 (1.272–1.734) *p* < 0.001
Age (All)	1.005 (1.001–1.009) *p* = 0.009				
<50	*	*	*	*	*
50–64	1.673 (1.370–2.043) *p* < 0.001	1.886 (1.530–2.324) *p* < 0.001	1.832 (1.487–2.256) *p* < 0.001	1.850 (1.502–2.280) *p* < 0.001	1.872 (1.519–2.307) *p* < 0.001
65–74	1.803 (1.459–2.226) *p* < 0.001	2.274 (1.807–2.861) *p* < 0.001	2.165 (1.722–2.722) *p* < 0.001	2.187 (1.740–2.749) *p* < 0.001	2.171 (1.728–2.729) *p* < 0.001
≥75	1.227 (0.985–1.530) *p* = 0.068	1.884 (1.470–2.414) *p* < 0.001	1.728 (1.352–2.210) *p* < 0.001	1.735 (1.358–2.217) *p* < 0.001	1.702 (1.334–2.172) *p* < 0.001
Ethnicity/Race					
Hispanic/Latino	1.329 (1.017–1.736) *p* = 0.037	1.411 (1.070–1.861) *p* = 0.015	1.402 (1.064–1.846) *p* = 0.016	1.425 (1.082–1.878) *p* = 0.012	1.440 (1.093–1.897) *p* = 0.010
Non-Hispanic Black	0.842 (0.633–1.120) *p* = 0.237	0.823 (0.614–1.103) *p* = 0.191	0.833 (0.622–1.116) *p* = 0.221	0.829 (0.619–1.111) *p* = 0.209	0.833 (0.622–1.115) *p* = 0.220
Non-Hispanic White	*	*	*	*	*
Asian	1.829 (1.275–2.622) *p* = 0.001	2.025 (1.400–2.929) *p* <0.001	1.955 (1.352–2.825) *p* <0.001	1.935 (1.340–2.795) *p* <0.001	1.915 (1.326–2.765) *p* <0.001
Other/Unknown	1.474 (1.100–1.976) *p* = 0.009	1.460 (1.082–1.970) *p* = 0.013	1.462 (1.084–1.971) *p* = 0.013	1.464 (1.086–1.975) *p* = 0.013	1.469 (1.089–1.981) *p* = 0.012
BMI < 25 (reference)	*	*			
25 ≤ BMI < 30	1.328 (1.101–1.601) *p* = 0.003	1.350 (1.115–1.635) *p* = 0.002			
30 ≤ BMI < 35	1.306 (1.055–1.615) *p* = 0.014	1.456 (1.168–1.816) *p* < 0.001			
35 ≤ BMI < 40	1.490 (1.150–1.931) *p* = 0.003	1.809 (1.381–2.370) *p* < 0.001			
BMI ≥ 30	1.289 (1.119–1.486) *p* < 0.001		1.480 (1.274–1.719) *p* < 0.001		
BMI ≥ 35	1.461 (1.233–1.731) *p* < 0.001			1.758 (1.470–2.104) *p* < 0.001	
BMI ≥ 40	1.707 (1.367–2.132) *p* < 0.001	2.827 (2.157–3.705) *p* < 0.001			2.122 (1.680–2.680) *p* < 0.001
Hypertension	0.971 (0.844–1.117) *p* = 0.681	1.026 (0.844–1.248) *p* = 0.793	1.060 (0.873–1.289) *p* = 0.556	1.054 (0.868–1.281) *p* = 0.594	1.059 (0.872–1.287) *p* = 0.561
Hyperlipidemia	0.884 (0.747–1.047) *p* = 0.154	0.861 (0.713–1.040) *p* = 0.120	0.853 (0.707–1.029) *p* = 0.097	0.857 (0.710–1.034) *p* = 0.108	0.876 (0.726–1.057) *p* = 0.167
Diabetes	0.851 (0.727–0.997) *p* = 0.046	0.836 (0.679–1.030) *p* = 0.093	0.827 (0.672–1.017) *p* = 0.072	0.822 (0.668–1.012) *p* = 0.064	0.818 (0.665–1.007) *p* = 0.058
Heart failure	0.760 (0.577–1.001) *p* = 0.051	0.867 (0.640–1.173) *p* = 0.354	0.878 (0.649–1.187) *p* = 0.396	0.864 (0.639–1.169) *p* = 0.342	0.860 (0.635–1.163) *p* = 0.327
CAD	0.707 (0.536–0.933) *p* = 0.014	0.710 (0.526–0.958) *p* = 0.025	0.708 (0.525–0.955) *p* = 0.024	0.716 (0.531–0.966) *p* = 0.029	0.714 (0.529–0.963) *p* = 0.027
PAD	0.681 (0.401–1.156) *p* = 0.155	0.853 (0.494–1.473) *p* = 0.568	0.839 (0.487–1.448) *p* = 0.530	0.838 (0.485–1.446) *p* = 0.525	0.839 (0.486–1.448) *p* = 0.529
A fib	0.764 (0.553–1.056) *p* = 0.103	0.826 (0.586–1.163) *p* = 0.274	0.841 (0.598–1.184) *p* = 0.321	0.841 (0.598–1.184) *p* = 0.321	0.850 (0.604–1.197) *p* = 0.352
Stroke	1.111 (0.962–1.282) *p* = 0.153	1.121 (0.967–1.300) *p* = 0.131	1.133 (0.978–1.314) *p* = 0.097	1.134 (0.978–1.315) *p* = 0.095	1.132 (0.977–1.313) *p* = 0.099
CKD	0.943 (0.757–1.173) *p* = 0.597	1.104 (0.870–1.400) *p* = 0.416	1.089 (0.859–1.381) *p* = 0.482	1.076 (0.848–1.364) *p* = 0.548	1.077 (0.850–1.366) *p* = 0.539
COPD	0.853 (0.444–1.636) *p* = 0.632	0.936 (0.479–1.827) *p* = 0.846	0.893 (0.458–1.740) *p* = 0.739	0.885 (0.454–1.726) *p* = 0.720	0.884 (0.453–1.722) *p* = 0.717
Asthma	0.934 (0.739–1.182) *p* = 0.571	0.957 (0.749–1.223) *p* = 0.726	1.006 (0.789–1.283) *p* = 0.963	0.980 (0.768–1.251) *p* = 0.871	0.973 (0.762–1.243) *p* = 0.828
HIV/AIDS	1.172 (0.762–1.803) *p* = 0.471	1.323 (0.852–2.055) *p* = 0.212	1.288 (0.830–1.998) *p* = 0.259	1.298 (0.837–2.013) *p* = 0.245	1.266 (0.816–1.963) *p* = 0.292
Liver cirrhosis	0.789 (0.425–1.467) *p* = 0.454	0.782 (0.416–1.468) *p* = 0.444	0.773 (0.412–1.451) *p* = 0.423	0.776 (0.414–1.456) *p* = 0.430	0.771 (0.411–1.446) *p* = 0.418

Notes: (1) BMI in kg/m^2^, (2) age in years, (3) Model A includes the variables that were found to have significant univariate association and BMI as ordinal variable: BMI < 25, 25 ≤ BMI < 30, 30 ≤ BMI < 35, 35 ≤ BMI < 40, and BMI ≥ 40, (4) Model B includes the variables that were found to have significant univariate association and BMI as dichotomous variable: BMI < 30 and BMI ≥ 30, (5) Model C includes the variables that were found to have significant univariate association and BMI as dichotomous variable: BMI < 35 and BMI ≥ 35, (6) Model D includes the variables that were found to have significant univariate association and BMI as dichotomous variable: BMI < 40 and BMI ≥ 40. Abbreviations: BMI = body mass index, OR = odds ratio, CI = confidence interval, kg = kilogram, m = meter, CAD = coronary artery disease, PAD = peripheral artery disease, Afib = atrial fibrillation, CKD = chronic kidney disease, COPD = chronic obstructive pulmonary disease, HIV = human immunodeficiency, AIDS = acquired immunodeficiency syndrome, * = reference group.

**Table 6 jcm-11-00622-t006:** Baseline patient characteristics: univariate and multivariate logistic regression analysis for the outcome of admission to the intensive care unit.

Variable	Univariate Analysis	Multivariate Analysis			
		Model A	Model B	Model C	Model D
		*n* = 8833	*n* = 8833	*n* = 8833	*n* = 8833
	OR, 95% CI, *p*-Value	OR, 95% CI, *p*-Value	OR, 95% CI, *p*-Value	OR, 95% CI, *p*-Value	OR, 95% CI, *p*-Value
Male sex	1.672 (1.497–1.867) *p* = <0.001	1.669 (1.486–1.875) *p* = <0.001	1.641 (1.462–1.843) *p* = <0.001	1.622 (1.445–1.820) *p* = <0.001	1.581 (1.410–1.773) *p* = <0.001
Age (All)	0.998 (0.995–1.001) *p* = 0.248				
<50	*	*	*	*	*
50–64	1.598 (1.389–1.837) *p* = <0.001	1.666 (1.437–1.931) *p* = <0.001	1.640 (1.415–1.901) *p* = <0.001	1.648 (1.422–1.910) *p* = <0.001	1.654 (1.427–1.917) *p* = <0.001
65–74	1.441 (1.238–1.679) *p* = <0.001	1.601 (1.354–1.893) *p* = <0.001	1.558 (1.318–1.842) *p* = <0.001	1.553 (1.314–1.835) *p* = <0.001	1.533 (1.298–1.811) *p* = <0.001
≥75	0.802 (0.682–0.944) *p* = 0.008	1.011 (0.840–1.216) *p* = 0.911	0.964 (0.802–1.159) *p* = 0.698	0.946 (0.788–1.135) *p* = 0.549	0.917 (0.764–1.100) *p* = 0.350
Ethnicity/Race					
Hispanic/Latino	1.127 (0.929–1.367) *p* = 0.225	1.066 (0.871–1.305) *p* = 0.537	1.061 (0.867–1.298) *p* = 0.565	1.075 (0.879–1.315) *p* = 0.480	1.082 (0.885–1.324) *p* = 0.441
Non-Hispanic Black	0.896 (0.733–1.096) *p* = 0.286	0.841 (0.682–1.036) *p* = 0.104	0.847 (0.688–1.044) *p* = 0.120	0.845 (0.686–1.041) *p* = 0.113	0.851 (0.691–1.049) *p* = 0.130
Non-Hispanic White	*	*	*	*	*
Asian	1.902 (1.456–2.484) *p* = <0.001	1.943 (1.475–2.561) *p* = <0.001	1.910 (1.450–2.516) *p* = <0.001	1.856 (1.410–2.443) *p* = <0.001	1.818 (1.382–2.393) *p* = <0.001
Other/Unknown	1.384 (1.119–1.712) *p* = 0.003	1.279 (1.026–1.595) *p* = 0.029	1.280 (1.027–1.595) *p* = 0.028	1.279 (1.026–1.593) *p* = 0.028	1.280 (1.027–1.594) *p* = 0.028
BMI < 25 (reference)	*	*			
25 ≤ BMI < 30	1.172 (1.022–1.343) *p* = 0.023	1.157 (1.005–1.332) *p* = 0.043			
30 ≤ BMI < 35	1.372 (1.178–1.598) *p* = <0.001	1.459 (1.244–1.711) *p* = <0.001			
35 ≤ BMI < 40	1.562 (1.293–1.885) *p* = <0.001	1.766 (1.449–2.152) *p* = <0.001			
BMI ≥ 30	1.378 (1.240–1.531) *p* = <0.001		1.525 (1.363–1.706) *p* = <0.001		
BMI ≥ 35	1.438 (1.263–1.636) *p* = <0.001			1.639 (1.429–1.880) *p* = <0.001	
BMI ≥ 40	1.496 (1.253–1.787) *p* = <0.001	2.194 (1.777–2.708) *p* = <0.001			1.742 (1.446–2.099) *p* = <0.001
Hypertension	0.967 (0.872–1.073) *p* = 0.531	1.002 (0.867–1.157) *p* = 0.980	1.021 (0.884–1.179) *p* = 0.780	1.021 (0.885–1.179) *p* = 0.773	1.031 (0.893–1.190) *p* = 0.677
Hyperlipidemia	0.926 (0.818–1.047) *p* = 0.219	0.916 (0.797–1.052) *p* = 0.215	0.912 (0.794–1.048) *p* = 0.193	0.918 (0.799–1.054) *p* = 0.225	0.933 (0.812–1.071) *p* = 0.324
Diabetes	0.931 (0.827–1.049) *p* = 0.242	0.976 (0.836–1.139) *p* = 0.758	0.970 (0.832–1.132) *p* = 0.702	0.964 (0.826–1.124) *p* = 0.640	0.960 (0.823–1.120) *p* = 0.605
Heart failure	0.802 (0.659–0.976) *p* = 0.028	0.808 (0.647–1.008) *p* = 0.059	0.815 (0.653–1.016) *p* = 0.069	0.806 (0.646–1.006) *p* = 0.056	0.806 (0.647–1.006) *p* = 0.056
CAD	0.939 (0.781–1.129) *p* = 0.503	0.979 (0.796–1.204) *p* = 0.839	0.976 (0.794–1.200) *p* = 0.818	0.983 (0.800–1.209) *p* = 0.874	0.978 (0.796–1.202) *p* = 0.832
PAD	0.757 (0.525–1.092) *p* = 0.136	0.860 (0.586–1.262) *p* = 0.439	0.851 (0.580–1.247) *p* = 0.407	0.845 (0.576–1.240) *p* = 0.390	0.848 (0.578–1.242) *p* = 0.397
A fib	1.271 (1.036–1.560) *p* = 0.022	1.562 (1.246–1.958) *p* = <0.001	1.576 (1.258–1.975) *p* = <0.001	1.584 (1.265–1.985) *p* = <0.001	1.599 (1.276–2.002) *p* = <0.001
Stroke	0.804 (0.720–0.897) *p* = <0.001	0.811 (0.724–0.909) *p* = <0.001	0.817 (0.729–0.915) *p* = <0.001	0.817 (0.729–0.915) *p* = <0.001	0.817 (0.729–0.916) *p* = <0.001
CKD	0.993 (0.846–1.164) *p* = 0.926	1.098 (0.920–1.309) *p* = 0.300	1.091 (0.915–1.300) *p* = 0.334	1.074 (0.901–1.281) *p* = 0.424	1.072 (0.899–1.278) *p* = 0.438
COPD	0.710 (0.428–1.176) *p* = 0.183	0.783 (0.466–1.317) *p* = 0.357	0.764 (0.455–1.284) *p* = 0.309	0.749 (0.446–1.259) *p* = 0.275	0.745 (0.444–1.251) *p* = 0.266
Asthma	0.792 (0.661–0.948) *p* = 0.011	0.799 (0.661–0.965) *p* = 0.020	0.822 (0.682–0.992) *p* = 0.041	0.811 (0.672–0.979) *p* = 0.029	0.818 (0.677–0.987) *p* = 0.036
HIV/AIDS	1.027 (0.734–1.437) *p* = 0.876	1.082 (0.766–1.527) *p* = 0.656	1.069 (0.758–1.508) *p* = 0.703	1.066 (0.756–1.503) *p* = 0.714	1.039 (0.737–1.464) *p* = 0.828
Liver cirrhosis	0.661 (0.410–1.064) *p* = 0.089	0.608 (0.374–0.988) *p* = 0.045	0.605 (0.372–0.982) *p* = 0.042	0.601 (0.370–0.976) *p* = 0.040	0.596 (0.367–0.969) *p* = 0.037

Notes: (1) BMI in kg/m^2^, (2) age in years, (3) Model A includes the variables that were found to have significant univariate association and BMI as ordinal variable: BMI < 25, 25 ≤ BMI < 30, 30 ≤ BMI < 35, 35 ≤ BMI < 40, and BMI ≥ 40, (4) Model B includes the variables that were found to have significant univariate association and BMI as dichotomous variable: BMI < 30 and BMI ≥ 30, (5) Model C includes the variables that were found to have significant univariate association and BMI as dichotomous variable: BMI < 35 and BMI ≥ 35, (6) Model D includes the variables that were found to have significant univariate association and BMI as dichotomous variable: BMI < 40 and BMI ≥ 40. Abbreviations: BMI = body mass index, OR = odds ratio, CI = confidence interval, kg = kilogram, m = meter, CAD = coronary artery disease, PAD = peripheral artery disease, Afib = atrial fibrillation, CKD = chronic kidney disease, COPD = chronic obstructive pulmonary disease, HIV = human immunodeficiency, AIDS = acquired immunodeficiency syndrome, * = reference group.

**Table 7 jcm-11-00622-t007:** BMI group: sex-based subgroup analysis of the multivariate logistic regression results for the outcome of mortality.

Variable	Sex	
	Female	Male
	*n* = 3593 (40.67%)	*n* = 5240 (59.33%)
	OR, 95% CI, *p*-Value	OR, 95% CI, *p*-Value
Age < 65	0.243 (0.199–0.295) *p* < 0.001	0.296 (0.256–0.343) *p* < 0.001
Age ≥ 65	4.121 (3.387–5.014) *p* < 0.001	3.376 (2.919–3.905) *p* < 0.001
BMI < 25 (reference)	*	*
25 ≤ BMI < 30	1.045 (0.837–1.306) *p* = 0.697	1.188 (1.011–1.397) *p* = 0.036
30 ≤ BMI < 35	1.057 (0.824–1.354) *p* = 0.664	1.188 (1.011–1.397) *p* < 0.001
35 ≤ BMI < 40	1.731 (1.302–2.301) *p* < 0.001	1.979 (1.522–2.573) *p* < 0.001
BMI ≥ 40	1.954 (1.445–2.643) *p* < 0.001	2.795 (2.098–3.722) *p* < 0.001

Notes: (1) N = 8833, (2) BMI in kg/m^2^, (3) age in years. Abbreviations: BMI = body mass index, OR = odds ratio, CI = confidence interval, kg = kilogram, m = meter, * = reference group.

**Table 8 jcm-11-00622-t008:** BMI group: age-based subgroup analysis of the multivariate logistic regression results for the outcome of mortality.

Variable	Age	
	Age < 65	Age ≥ 65
	*n* = 4871 (55.15%)	*n* = 3962 (44.85%)
	OR, 95% CI, *p*-Value	OR, 95% CI, *p*-Value
Male	1.818 (1.520–2.175) *p* < 0.001	1.219 (1.062–1.398) *p*= 0.005
Female	0.550 (0.460–0.658) *p* = <0.001	0.821 (0.715–0.941) *p* = 0.005
BMI < 25 (reference)	*	*
25 ≤ BMI < 30	1.240 (0.976–1.575) *p* = 0.078	1.144 (0.976–1.341) *p* = 0.098
30 ≤ BMI < 35	1.670 (1.294–2.154) *p* < 0.001	1.236 (1.014–1.507) *p* = 0.036
35 ≤ BMI < 40	2.337 (1.750–3.121) *p* < 0.001	1.634 (1.245–2.145) *p <* 0.001
BMI ≥ 40	2.943 (2.188–3.957) *p* < 0.001	1.959 (1.436–2.674) *p* < 0.001

Notes: (1) N = 8833, (2) BMI in kg/m2, (3) age in years. Abbreviations: BMI = body mass index, OR = odds ratio, CI = confidence interval, kg = kilogram, m = meter, * = reference group.

**Table 9 jcm-11-00622-t009:** BMI group: subgroup analysis of the multivariate logistic regression results for the outcome of mortality based on the ethnic or racial background.

Variable	Race/Ethnicity				
	Non-Hispanic Black	Hispanic/Latino	Non-Hispanic White	Asian	Other/Unknown
	*n* = 2702 (30.59%)	*n* = 3437 (38.91%)	*n* = 832 (9.42%)	*n* = 445 (5.04%)	*n* = 1417 (16.04%)
	OR, 95% CI, *p*-Value	OR, 95% CI, *p*-Value	OR, 95% CI, *p*-Value	OR, 95% CI, *p*-Value	OR, 95% CI, *p*-Value
Male	1.454 (1.196–1.766) *p* < 0.001	1.526 (1.271–1.832) *p* < 0.001	1.099 (0.794–1.522) *p* = 0.568	2.493 (1.522–4.083) *p* < 0.001	1.367 (1.047–1.785) *p* = 0.022
Female	0.688 (0.566–0.836) *p* < 0.001	0.655 (0.546–0.787) *p* = <0.001	0.910 (0.657–1.259) *p* = 0.568	0.401 (0.245–0.657) *p* < 0.001	0.732 (0.560–0.955) *p* = 0.022
Age < 65	0.339 (0.275–0.418) *p* < 0.001	0.231 (0.190–0.281) *p* = <0.001	0.233 (0.154–0.352) *p* < 0.001	0.270 (0.163–0.447) *p* < 0.001	0.306 (0.235–0.400) *p* < 0.001
Age ≥ 65	2.950 (2.391–3.640) *p* < 0.001	4.334 (3.563–5.271) *p* < 0.001	4.295 (2.842–6.491) *p* < 0.001	3.701 (2.238–6.119) *p* < 0.001	3.263 (2.499–4.262) *p* < 0.001
BMI < 25 (reference)	*	*	*	*	*
25 ≤ BMI < 30	1.184 (0.927–1.513) *p* = 0.177	1.308 (1.042–1.643) *p* = 0.021	0.875 (0.593–1.290) *p* = 0.499	1.102 (0.652–1.863) *p* = 0.716	1.116 (0.816–1.525) *p* = 0.492
30 ≤ BMI < 35	1.196 (0.900–1.590) *p* = 0.218	1.623 (1.258–2.093) *p* < 0.001	0.978 (0.609–1.573) *p* = 0.928	1.838 (0.852–3.966) *p* = 0.121	1.377 (0.951–1.994) *p* = 0.090
35 ≤ BMI < 40	1.765 (1.261–2.469) *p* < 0.001	2.103 (1.522–2.905) *p* < 0.001	1.835 (0.975–3.453) *p* = 0.060	1.349 (0.398–4.567) *p* = 0.631	2.047 (1.297–3.231) *p* = 0.002
BMI ≥ 40	2.068 (1.463–2.923) *p* < 0.001	2.622 (1.815–3.789) *p* < 0.001	2.474 (1.295–4.726) *p* = 0.006	9.535 (2.113–43.033) *p* = 0.003	2.470 (1.519–4.018) *p* < 0.001

Notes: (1) N = 8833, (2) BMI in kg/m^2^, (3) age in years. Abbreviations: BMI = body mass index, OR = odds ratio, CI = confidence interval, kg = kilogram, m = meter, * = reference group.

**Table 10 jcm-11-00622-t010:** BMI group: subgroup analysis of the multivariate logistic regression results for the outcome of mortality based on the timing of hospitalization.

Variable	Quartile			
	First Quartile (3/3–29/3/2020)	Second Quartile (29/3–8/4/2020)	Third Quartile (8/4–23/4/2020)	Last Quartile (23/4–31/10/2020)
	*n* = 2209 (25.01%)	*n* = 2208 (25.00%)	*n* = 2208 (25.00%)	*n* = 2208 (25.00%)
	OR, 95% CI, *p*-Value	OR, 95% CI, *p*-Value	OR, 95% CI, *p*-Value	OR, 95% CI, *p*-Value
Male	1.688 (1.361–2.094) *p* < 0.001	1.287 (1.056–1.568) *p* = 0.012	1.388 (1.131–1.703) *p* = 0.002	1.010 (0.741–1.375) *p* = 0.951
Female	0.592 (0.478–0.735) *p* < 0.001	0.777 (0.638–0.947) *p* = 0.012	0.720 (0.587–0.884) *p* = 0.002	0.990 (0.727–1.349) *p* = 0.951
Age < 65	0.268 (0.215–0.335) *p* < 0.001	0.321 (0.261–0.394) *p* < 0.001	0.278 (0.222–0.348) *p* < 0.001	0.237 (0.165–0.339) *p* < 0.001
Age ≥ 65	3.728 (2.987–4.654) *p* < 0.001	3.116 (2.536–3.830) *p* < 0.001	3.603 (2.878–4.510) *p* < 0.001	4.228 (2.953–6.053) *p* < 0.001
BMI < 25 (reference)	*	*	*	*
25 ≤ BMI < 30	1.178 (0.900–1.542) *p* = 0.234	1.031 (0.813–1.307) *p* = 0.800	1.113 (0.873–1.421) *p* = 0.387	0.869 (0.594–1.270) *p* = 0.468
30 ≤ BMI < 35	1.443 (1.071–1.946) *p* = 0.016	1.115 (0.839–1.481) *p* = 0.455	1.339 (1.001–1.791) *p* = 0.049	1.024 (0.659–1.591) *p* = 0.916
35 ≤ BMI < 40	2.102 (1.443–3.061) *p* < 0.001	1.460 (1.045–2.039) *p* = 0.027	1.536 (1.049–2.249) *p* = 0.027	1.594 (0.920–2.762) *p* = 0.096
BMI ≥ 40	3.148 (2.160–4.588) *p* < 0.001	1.941 (1.319–2.855) *p* < 0.001	1.598 (1.068–2.392) *p* = 0.023	1.339 (0.683–2.628) *p* = 0.396

Notes: (1) N = 8833, (2) BMI in kg/m^2^, (3) age in years; (4) for this analysis, the total cohort of 8833 patients was divided in four equal quartile cohorts based on the day of admission (1st quartile: 3/3–29/3/2020, 2nd quartile: 29/3–8/4/2020, 3rd quartile: 8/4–23/4/2020, 4th quartile: 23/4–31/10/2020). Abbreviations: BMI = body mass index, OR = odds ratio, CI = confidence interval, kg = kilogram, m = meter, * = reference group.

**Table 11 jcm-11-00622-t011:** Inflammatory markers: univariate and multivariate logistic regression analysis for the outcome of mortality.

Variable	No.	Univariate Analysis	Multivariate Analysis		
			Model A	Model B	Model C
			*n* = 5068	*n* = 2461	*n* = 665
		OR, 95% CI, *p*-Value	OR, 95% CI, *p*-Value	OR, 95% CI, *p*-Value	OR, 95% CI, *p*-Value
Male sex		1.192 (1.081–1.315) *p* < 0.001	1.320 (1.142–1.526) *p* < 0.001	1.178 (0.952–1.457) *p* = 0.132	1.085 (0.739–1.595) *p* = 0.677
Age (All)		1.045 (1.042–1.048) *p* < 0.001			
<50 (reference)		1.067 (1.046–1.088) *p* < 0.001	*	*	*
50–64		1.043 (1.020–1.067) *p* < 0.001	1.973 (1.589–2.450) *p* < 0.001	2.132 (1.552–2.929) *p* < 0.001	2.743 (1.615–4.658) *p* < 0.001
65–74		1.066 (1.029–1.104) *p* < 0.001	3.146 (2.514–3.937) *p* < 0.001	3.571 (2.567–4.970) *p* < 0.001	3.635 (2.051–6.443) *p* < 0.001
≥75		1.034 (1.019–1.049) *p* < 0.001	5.204 (4.119–6.575) *p* < 0.001	5.616 (3.970–7.943) *p* < 0.001	6.109 (3.288–11.349) *p* < 0.001
BMI < 25 (reference)		0.944 (0.850–1.049) *p* = 0.287	*	*	*
25 ≤ BMI < 30		0.947 (0.856–1.049) *p* = 0.297	1.386 (1.162–1.653) *p* < 0.001	1.261 (0.972–1.636) *p* = 0.081	1.343 (0.823–2.193) *p* = 0.238
30 ≤ BMI < 35		0.927 (0.822–1.047) *p* = 0.223	1.583 (1.290–1.943) *p* < 0.001	1.606 (1.190–2.167) *p* = 0.002	2.382 (1.383–4.104) *p* = 0.002
35 ≤ BMI < 40		1.163 (0.992–1.363) *p* = 0.063	2.010 (1.567–2.578) *p* < 0.001	2.098 (1.473–2.990) *p* < 0.001	3.651 (1.928–6.913) *p* < 0.001
BMI ≥ 40		1.346 (1.136–1.594) *p* < 0.001	3.055 (2.335–3.996) *p* < 0.001	2.754 (1.893–4.008) *p* < 0.001	3.280 (1.722–6.248) *p* < 0.001
WBC—10^3^/μL	*n* = 8795	1.045 (1.036–1.055) *p* < 0.001	0.848 (0.677–1.061) *p* = 0.149	0.786 (0.578–1.069) *p* = 0.125	0.853 (0.521–1.396) *p* = 0.0.527
Neutrophils—10^3^/μL	*n* = 8795	1.074 (1.063–1.086) *p* < 0.001	1.234 (0.977–1.558) *p* = 0.077	1.321 (0.959–1.819) *p* = 0.088	1.206 (0.724–2.008) *p* = 0.472
Monocytes—10^3^/μL	*n* = 8795	0.748 (0.650–0.861) *p* < 0.001	0.695 (0.498–0.970) *p* = 0.032	0.852 (0.544–1.335) *p* = 0.486	0.918 (0.413–2.041) *p* = 0.834
Lymphocytes—10^3^/μL	*n* = 8795	0.990 (0.967–1.014) *p* = 0.419	1.208 (0.962–1.516) *p* = 0.104	1.311 (0.958–1.793) *p* = 0.090	1.202 (0.727–1.986) *p* = 0.474
LDH per 100 U/L	*n* = 6476	1.003 (1.002–1.003) *p* < 0.001	1.209 (1.178–1.241) *p* < 0.001	1.191 (1.148–1.236) *p* < 0.001	1.189 (1.108–1.276) *p* < 0.001
Ferritin per 200 ng/mL	*n* = 6285	1.000 (1.000–1.000) *p* < 0.001	1.009 (1.002–1.017) *p* = 0.009	1.012 (1.002–1.021) *p* = 0.015	1.037 (1.014–1.062) *p* = 0.002
Albumin—g/dL	*n* = 8357	0.408 (0.373–0.445) *p* < 0.001	0.591 (0.518–0.673) *p* < 0.001	0.530 (0.438–0.641) *p* < 0.001	0.696 (0.492–0.984) *p* = 0.040
D-dimer per 200 mg/mL	*n* = 6471	1.008 (1.007–1.009) *p* < 0.001	1.003 (1.002–1.004) *p* < 0.001	1.003 (1.001–1.005) *p* = 0.004	1.002 (0.999–1.006) *p* = 0.194
CRP per 20 mg/dL	*n* = 3114	1.063 (1.046–1.080) *p* < 0.001		1.034 (1.013–1.055) *p* = 0.002	1.018 (0.978–1.060) *p* = 0.385
IL-6 per 20 pg/mL	*n* = 2237	1.020 (1.015–1.025) *p* < 0.001			1.011 (1.003–1.019) *p* = 0.007
N/R	*n* = 8795	1.051 (1.044–1.058) *p* < 0.001	*	*	*
L/CRP	*n* = 3109	0.639 (0.532–0.783) *p* < 0.001	*	*	*

Notes: (1) BMI in kg/m^2^, (2) age in years, (3) Model A refers to a cohort of 5068 patients that had available results for all these markers: WBC, neutrophils, monocytes, lymphocytes, LDH, ferritin, albumin, d-dimer, (4) Model B refers to a cohort of 2461 patients that had available results for all the markers of Model A and CRP, (5) Model C refers to a cohort of 665 patients that had available results for all the markers of Model B and IL-6, (6) neutrophils, lymphocytes, and CRP were included in the multivariate analysis; therefore, N/L and L/CRP were not included. Abbreviations: BMI = body mass index, g = gram, ng = nanogram, mg = milligram, L = liter, dL = deciliter, mL = milliliter, μL = microliter, U = unit, CRP = C-reactive protein, IL-6 = interleukin-6, LDH = lactic dehydrogenase, N/L = neutrophil-lymphocyte ratio, L/CRP = lymphocyte—CRP ratio, * = reference group.

## Data Availability

All relevant data are within the manuscript and its supporting information files.

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
