# Peer review of "Obesity, Inflammation, and Mortality in COVID-19: An Observational Study from the Public Health Care System of New York City"

_jcm, 2022, doi:10.3390/jcm11030622_

Round 1
Reviewer 1 Report
The subject matter discussed in the manuscript is interesting with excellent population size, however, most of the features discussed here are known to the field. The authors presented their great knowledge and scientific skills. I do not see any significant shortcomings.
My only suggestion is to improve the quality of the Figures, as some descriptions are difficult to read. Some method sections require details like how cytokines and ferritin levels were measured.
Author Response
The quality of the figures has been improved and uploaded.
Labs: All laboratory tests refer to the first available measured within 24 hours from admission.
1. D-dimer: ACL TOP 550 Instrument Laboratory, assay Latex Immuno Turbidometric quantitative
2. LDH: Roche Diagnostics 701, enzymatic: LDH catalyzes the conversion of L-lactate to pyruvate; NAD is reduced to NADH in the process
3. Ferritin: Roche Diagnostics e801, assay: Electrochemiluminescence Immunoassay (ECLIA) monoclonal ferritin‑specific antibody
4. CRP: Roche Diagnostics e502, assay: determined turbidimetrically, Particle enhanced immunoturbidimetric assay
5. IL-6: Elecsys IL-6–Roche Diagnostics
Reviewer 2 Report
The paper of Palaiodimos et al. entitled Obesity and Mortality in Hospitalized Patients with COVID-19, Body Mass Index and Inflammation: An Observational Study of 8833 Patients from the Public Health Care System of New York City found overweight and obesity independently associated with in-hospital mortality, whereas obesity was not associated with higher inflammatory markers in the study cohort. I read the paper with great interest. In literature, obesity and other comorbidities (e.g., diabetes, hypertension) are known to be associated severe COVID-19. So, Palaiodimos et al. thus also provide an important contribution to further understanding about COVID-19 and obesity, and thus for attention to this patient group that is important in therapy.
In the present study, Palaiodimos et al. found in a large patient cohort overweight and obesity associated with in-hospital mortality, invasive mechanical ventilation, and admission to ICU. For data analyses BMI, sex, ages, race, ethnicity, co-morbidities, and clinical parameters (relevant for COVID-19) were included. Based on the results the study group found overweight and obesity associated to in-hospital mortality, but not to inflammatory markers. Some remarks on the present manuscript must nevertheless be put forward for discussion in order to give the reader more clarity.
First, my suggestion would be to make the title clearer to better convey the importance of the topic to the reader.
The severity of illness by ICU admission and invasive mechanical ventilation was also included in the analysis. What about other treatment options, especially ECMO?
Interestingly, the evaluation showed no negative effects in diseases such as diabetes or hypertension and COVID-19 in this patient group, which is known to often suffer from these co-morbidities due to obesity. Are there any explanations for this?
I don't quite understand the definition time of admission to hospital in terms of the defined quartiles. According to what was this defined.
Certainly, the discussed causes for differences to mortality observed at later time of admission play a significant role. Are data also collected beyond October 2020 that show this?
An association between higher inflammatory markers and obesity was not found. Were the inflammation markers different between the groups during disease progression, e.g., before admission to ICU or before invasive mechanical ventilation? Did the inflammation markers indicate disease progression in the groups with overweight or obesity in addition to the differences found? The role of cytokine release in COVID-19 and its impact is controversy discussed in literature, also when compared to other cytokine releasing disease such as ARDS, Sepsis, CRS. Further, immune tolerance might be a player in situations of chronical immune stress? (PMID: 29716792)
Based on scientific data and experience over nearly 2 years of pandemic, vaccination should be recommended for all adults.
Author Response
- First, my suggestion would be to make the title clearer to better convey the importance of the topic to the reader.
*Thank you for this comment. We simplified the title to:
“Obesity, Inflammation, and Mortality in COVID-19: an observational study from the public health care system of New York City”
- The severity of illness by ICU admission and invasive mechanical ventilation was also included in the analysis. What about other treatment options, especially ECMO?
*Thank you for this comment. We had decided to focus more on death/mortality regarding outcomes acknowledging that the risk of bias would be higher with other outcomes, e.g. decisions around intubation or decisions around ICU admission. Regarding ECMO, unfortunately the public health care system in NYC does not routinely practice ECMO. Only one out of eleven hospitals were capable of practicing ECMO at that time (Bellevue Hospital), however its practice was limited. To the best of our knowledge, ECMO was not used significantly during the study period, that mainly covers the first and really intense wave of the pandemic (spring 2020).
- Interestingly, the evaluation showed no negative effects in diseases such as diabetes or hypertension and COVID-19 in this patient group, which is known to often suffer from these co-morbidities due to obesity. Are there any explanations for this?
*This is another excellent comment. We have been perplexed about the findings about hypertension and diabetes. This finding is not unique to our cohort. There are many cohorts with mixed results about this comorbidity. This review https://www.ncbi.nlm.nih.gov/pmc/articles/PMC7925389/ summarized some of these mixed results. One of our explanations is that there is tremendous heterogeneity among patients with hypertension or diabetes; recent or chronic, controlled or uncontrolled. These patients might also have taken different regimen or not at all. For example, there is growing literature showing that statins might be protective and patients with hypertension or diabetes are more likely to take statins (although the results are mixed in the statins topic, as well). This is an excellent topic for a separate study with a different objective. Unless you instruct otherwise, we would prefer not to expand further and strictly focus on obesity/BMI.
- I don't quite understand the definition time of admission to hospital in terms of the defined quartiles. According to what was this defined.
*Thank you for giving us the opportunity to clarify. Indeed, clarification is needed. We added this line as an additional note in the respective table (Table 10) and a similar note on Table 4:
“For this analysis, the total cohort of 8833 patients was divided in four equal quartile cohorts based on the day of admission (1stquartile: 3/3-3/29/2020, 2nd quartile: 3/29-4/8/2020, 3rd quartile: 4/8-4/23/2020, 4th quartile: 4/23-10/31/2020)”
Similarly, the paragraph above the table presenting this analysis reads
“A subgroup analysis for the timing of hospitalization was performed after the cohort was divided in four patient quartiles based on the day of admission (3/3-3/29/2020, 3/29-4/8/2020, 4/8-4/23/2020, 4/23-10/31/2020)”
This was an arbitrary way that we predefined to show whether death rate and associations of interest changed over time.
- Certainly, the discussed causes for differences to mortality observed at later time of admission play a significant role. Are data also collected beyond October 2020 that show this?
*Thank you for this comment. Unfortunately, we do not have data after October 31st, 2020. Our IRB approval was extended up to the end of October 2020. The positive thing is that we have had very low number of hospital admissions in our system and in NYC from November 2020 to a few weeks ago when Omicron variant arrived. Although we expect that the death rate continued to decline by the time, we do not have the necessary data to demonstrate it after October 2020.
- An association between higher inflammatory markers and obesity was not found. Were the inflammation markers different between the groups during disease progression, e.g., before admission to ICU or before invasive mechanical ventilation? Did the inflammation markers indicate disease progression in the groups with overweight or obesity in addition to the differences found? The role of cytokine release in COVID-19 and its impact is controversy discussed in literature, also when compared to other cytokine releasing disease such as ARDS, Sepsis, CRS. Further, immune tolerance might be a player in
These are very interesting thoughts. Thank you very much. We did not have a significant number of follow-up inflammatory markers in our population at the study period that was mainly the first wave of the pandemic in NYC. A study from another institution revealed that patients that did not have > 50% decline in CRP within 2-3 days had higher likelihood for death. This finding was present across the broad with regards to BMI (DOI: 10.12788/jhm.3560).
- Based on scientific data and experience over nearly 2 years of pandemic, vaccination should be recommended for all adults.
*Thank you for your excellent comment. We modified a section of the concluding paragraph to read:
“Not only patients with severe obesity but also those with overweight should be highly encouraged to receive full vaccination regimen against SARS-CoV-2 and they should be prioritized to receive neutralizing antibodies or antivirals as early treatments”
Round 2
Reviewer 2 Report
Thanks Palaiodimos et al. for the revised manuscript. There are no further comments.